# Screening People with Tuberculosis for High Risk of Severe Illness at Notification: Programmatic Experience from Karnataka, India

**DOI:** 10.3390/tropicalmed6020102

**Published:** 2021-06-15

**Authors:** Hemant Deepak Shewade, Sharath Burugina Nagaraja, Hosadurga Jagadish Deepak Murthy, Basavarajachar Vanitha, Madhavi Bhargava, Anil Singarajipura, Suresh G. Shastri, Ramesh Chandra Reddy, Ajay M. V. Kumar, Anurag Bhargava

**Affiliations:** 1International Union Against Tuberculosis and Lung Disease (The Union), 75006 Paris, France; akumar@theunion.org; 2The Union South-East Asia Office, New Delhi 110016, India; 3Employees’ State Insurance Corporation Medical College and PGIMSR, Bengaluru 560010, India; sharathbn@yahoo.com; 4Bangalore Medical College and Research Institute, Bengaluru 560002, India; drdeepakmhj@gmail.com; 5Bowring and Lady Curzon Medical College and Research Institute, Bengaluru 560001, India; vanisonu1510@gmail.com; 6Centre for Nutrition Studies, Yenepoya (Deemed to be University), Mangaluru 575018, India; madhavibhargava4@gmail.com (M.B.); anuragb17@gmail.com (A.B.); 7Yenepoya Medical College, Yenepoya (Deemed to be University), Mangaluru 575018, India; 8Department of Health and Family Welfare, Government of Karnataka, Bengaluru 560023, India; dadranil@gmail.com (A.S.); susha007@gmail.com (S.G.S.); vrcreddy@live.com (R.C.R.)

**Keywords:** TB mortality, people with TB who are severely ill, operational research, coverage, feasibility, mobile application, hospitalization, domiciliary care, programme settings

## Abstract

Due to limited availability of diagnostics and capacity, people with tuberculosis do not always undergo systematic assessment for severe illness (requiring inpatient care). In Karnataka (south India), para-medical programme staff used a screening tool to identify people at ‘high risk of severe illness’, defined using indicators of very severe undernutrition, abnormal vital signs and poor performance status (any one): (i) body mass index (BMI) ≤ 14.0 kg/m^2^ (ii) BMI ≤ 16.0 kg/m^2^ with bilateral leg swelling (iii) respiratory rate > 24/min (iv) oxygen saturation < 94% (v) inability to stand without support. Of 3020 adults notified from public facilities (15 October to 30 November 2020) in 16 districts, 1531 (51%) were screened (district-wise range: 13–90%) and of them, 538 (35%) were classified as ‘high risk of severe illness’. Short median delays in screening from notification (five days), and all five indicators being collected for 88% of patients, suggests the feasibility of using this tool in programme settings. However, districts with poor screening coverage require further attention. To end tuberculosis deaths, screening should be followed by referral to higher facilities for comprehensive clinical evaluation, to assess the need for inpatient care. Future studies should assess the validity (especially sensitivity in picking severely ill patients) of this screening tool.

## 1. Introduction

Globally, tuberculosis (TB) is one of the leading killer due to a single infectious agent. In 2019, there were an estimated 1.4 million TB-related deaths, with an estimated case fatality ratio of 14:100 [1]. The End TB targets by the World Health Organization include 90% reduction of TB deaths by 2030 and 95% reduction by 2035 (when compared to 2015) [2,3]. The targets for reduction of death are more immediate and ambitious than the targets for reduction of TB incidence.

Most of the deaths in the high TB burden countries occur ‘early’, within two months of treatment initiation and are primarily due to severe illness and late presentation [4,5,6,7,8,9,10]. These can be averted by detecting the missing 2.7 million people with TB (including drug-resistant forms and those with co-morbidities) as early as possible and treating them appropriately [11]. Among notified patients with TB, those at higher risk of death (people with TB who are severely ill) should be identified early and provided appropriate timely inpatient clinical care [12]. In any potentially fatal illness, an assessment of severity is essential. However, this has not been systematically done for patients with TB.

Globally, there is limited literature on the burden of severe illness at notification and the feasibility of collecting this data in routine programme settings. Like elsewhere, India’s National TB Elimination Programme (NTEP) does not assess and capture severity of illness, though certain indicators like weight, height (often not recorded, making assessment of body mass index not possible) and sputum smear microscopy grading are collected [13]. In 2017, the NTEP released guidance on the criteria for identifying severely ill patients with TB who require inpatient care (see Box 1) [14]. The 2021 NTEP guidance (released after completion of this study) categorically recommends severity assessment for all people with TB as soon as possible after diagnosis and referral for inpatient care, if severely ill [15].

Box 1Proposed criteria for inpatient care for people with TB based on nutritional and clinical assessment with a goal to reduce TB mortality in India (2017) [14].
**Presence of any one of the following:**
1.BMI < 14.0 kg/m^2^2.MUAC < 16.0 cm (if unable to stand for a measurement of weight and height)3.BMI 14.0–15.9 kg/m^2^ AND (bilateral pedal oedema OR inability to stand without support OR no appetite)4.MUAC 16.0–18.9 cm AND (bilateral pedal oedema OR inability to stand without support OR no appetite)5.Severe anaemia (Hb < 7 g/dL) with or without heart failure6.Unstable vital signs–pulse rate > 100 per minute OR RR > 24 per minute OR oxygen saturation < 94% OR systolic blood pressure < 100 mm Hg OR poor performance status (bed-ridden or extremely limited mobility)7.Complications of PTB–Example, moderate–massive haemoptysis, hydro-pneumothorax8.Complications of EPTB–Example, altered consciousness, seizures, lower limb weakness, suspected intestinal obstruction or perforation9.Complications to anti-TB treatment–drug induced hepatotoxicity or seizures10.Patients with comorbidities who need inpatient care to manage these comorbidities according to the judgement of the treating physician–Example, DM, HIV, liver or renal disease, alcohol addiction, tobacco addiction
TB—tuberculosis, BMI—body mass index (kg/m^2^), MUAC—mid upper arm circumference (cm), Hb—haemoglobin, RR—respiratory rate, PTB—pulmonary TB, EPTB—extra pulmonary TB, DM—diabetes mellitus, HIV—human immunodeficiency virus.

Similar to the tools used from developed countries [16,17], the 2017 guidance for inpatient care requires clinical, laboratory and radiological evaluation [14]. In the current context, it may be challenging to apply these criteria fully to all people with TB in routine programme settings [12]. The severity of illness could be screened indirectly by evaluation of vital signs, degree of wasting (reflected by body mass index—BMI) or performance status. A screening tool that is simple and easy to use and interpret is desirable. Para-medical TB programme and general health staff at peripheral health institutions (PHIs) with limited access to diagnostics should be able to use this and it should also be possible for use in community settings. A screening tool comprising five indicators has been proposed (see Box 2) [12]. Those screened positive using this tool (presence of any one indicator) may be considered as ‘high risk of severe illness’ and referred to higher facilities for comprehensive clinical evaluation and inpatient care, if eligible (see Box 2).

Box 2Tool to screen for ‘high risk of severe illness’ at notification among adults (≥15 years) with TB (without known drug-resistant disease at diagnosis) from public health facilities of 16 districts in Karnataka, India, 15 October to 30 November 2020 [12].
**If at least one of the following is present, then the person with TB is ‘high risk of severe illness (requires referral for clinical evaluation and assessment for inpatient care)**
1.Body mass index (BMI) less than or equal to (≥) 14.0 kg/m^2#^  (OR)2.BMI less than or equal to (≥) 16.0 kg/m^2^ with leg swelling^#^  (OR)3.Respiratory rate more than (>) 24 per minute^##^ (OR)4.Oxygen saturation less than (<) 94% ^##^  (OR)5.Not able to stand without support (standing with support/squatting/sitting/bed ridden)
TB—tuberculosis; # very severe undernutrition indicators; ## respirator insufficiency indicators.

We conducted an operational study in Karnataka (India), to assess the feasibility of using this screening tool by para-medical TB programme staff in routine programme settings and to estimate the burden of ‘high risk of severe illness’ among people with TB. To explore the possibility of further simplification of the tool, we also assessed the contribution of very severe undernutrition, respiratory insufficiency and inability to stand without support (performance status) indicators towards ‘high risk of severe illness’.

## 2. Materials and Methods

### 2.1. Study Design and Participants

This was a cross-sectional study involving primary as well as secondary data. Study participants included all adults (≥15 years) with TB (without known drug-resistant disease at diagnosis) notified by public PHIs in select districts (n = 16) of Karnataka (India) between 15 October and 30 November 2020. We conveniently selected the following study districts (in alphabetical order): Belgaum, Bellary, Bengaluru City, Bengaluru Rural, Bengaluru Urban, Chikballapur, Chikkamagaluru, Dakshina Kannada, Dharwad, Gulbarga, Hassan, Kolar, Ramanagara, Shivamogga, Tumakuru and Udupi. We included study participants irrespective of their treatment initiation status or transfer out status. We excluded patients transferred-in from other non-study districts.

### 2.2. Setting

India has the highest TB burden with an estimated case fatality ratio of 17:100 [1]. India has an ambitious plan to attain the 2030 WHO End TB targets by 2025 [18]. The COVID-19 response related lockdown has possibly contributed towards a 21% increase in estimated TB deaths in 2020. This might reverse the gains made over the past five years [19].

Karnataka is a state in south India with a population of ≈64 million. In 2019–20, the proportion of adults (15–49) with undernutrition (BMI < 18.5 kg/m^2^) varied by gender and residence: 12.9% (women, urban), 19.9% (women, rural), 11.5% (men, urban), 16.2% (men, rural) [20]. The state’s TB case notification rate in 2019 was 136 per 100,000 population (107 public and 29 private notified per 100,000 population) [21]. The treatment success rate of the 2018 cohort was 81%. The case fatality was 7%, one of the highest reported by a state in India [21].

The state NTEP infrastructure includes 31 districts, sub-district level administrative TB units and PHIs (public and private) with at least one medical doctor and designated microscopy centres (DMC) for sputum microscopy. Each DMC has a laboratory technician. A major shift under the national strategic plan (2017–25) is to notify and initiate every diagnosed patient on treatment at the site of diagnosis [18]. Paper-based registers maintained at each of these PHI has a line-list of patients notified, their management, and the treatment outcomes. Each sub-district level unit has a senior TB treatment supervisor (STS) who updates these details in the NIKSHAY application (a case-based, web-based electronic TB information management system) on his/her mobile tablet [22]. Each district under the NTEP has a dedicated data entry operator (DEO) and the public PHIs may have additional staff to support the STS, called TB-health visitor in the urban areas. Patients receive daily treatment under direct observation of a health care provider, community volunteer or a family member.

In the study districts during the study period, there was 39% under-notification of TB from public PHIs as compared to 2019 [22]. This was due to the shifting of laboratory technicians from periphery to higher facilities for COVID-19 testing. TB diagnosis was mostly available in the sub-district and district level facilities, which also had facilities for rapid molecular tests.

### 2.3. Screening Tool

The indicators used in the screening tool (see Box 2) are also included in the criteria for inpatient care (see Box 1). In the criteria for inpatient care (see Box 1), BMI has a ‘<’ sign [14], while our screening tool uses ‘≤’ sign. This was done to ensure that we do not miss any cases at the periphery due to errors in rounding off. Very severe undernutrition, abnormal vital signs and poor performance status are known risk factors for death and have a strong association with TB mortality [14,23,24,25,26]. Due to the COVID-19 pandemic, most of the public PHIs had a portable pulse oximeter to screen for hypoxia among people with COVID-19.

### 2.4. Data Collection, Variables and Sources of Data

Screening was carried out in routine programme settings by TB programme staff. Due to the COVID-19 related travel restrictions, the investigators conducted online training of DEOs and STS. The laboratory technicians and TB-health visitors at public PHIs were trained by the DEO/STS. All the public PHIs, STS and DEO were also provided with standard operating procedures for screening (in the form of a document and short videos) (see Appendix A).

The laboratory technician of DMC or the TB-health visitor collected the data in a paper-based form (see Appendix A). They were encouraged to take support from the staff nurse or medical doctor at the PHI. If the opportunity at the time of diagnosis was missed, then screening was done during baseline assessment for human immunodeficiency virus (HIV) and diabetes mellitus or at any other earliest opportunity. The STS transcribed the screening related details in the EpiCollect5 mobile application (an open access application that allows offline mobile or tablet-based data capture and synchronises data in cloud). The TB programme staff were expected to calculate BMI and fill in the details. The DEOs monitored the completeness (of those eligible, how many were screened) and correctness of screening related data (errors in data collection and BMI calculation) by referring to the EpiCollect5 web portal and then provided feedback. Before the study period, this data collection process was piloted for at least 10 days in all study districts to address any unforeseen issues.

### 2.5. Data Management

On 15 December 2020, data were extracted from NIKSHAY (routinely captured secondary data) and EpiCollect5 (primary data) in Microsoft Excel^®^ 2010 (Microsoft, Redmond, WA, USA) and merged using the unique NIKSHAY identifier (NIKSHAY–parent database). We analyzed the data using EpiData Analysis (v2.2.2.183 EpiData Association, Odense, Denmark) and STATA (v12.1, copyright 1985–2011, Stata Corp. LP College Station, TX, USA) software.

For every study participant in the NIKSHAY database, we checked to establish if screening details had been captured in the EpiCollect5 database using the NIKSHAY identifier. Accordingly, we classified screening as ‘yes’ or ‘no’. We also calculated the BMI and ‘high risk of severe illness’ status using criteria mentioned in Box 2. We interpreted an error in BMI calculation (‘yes’) if the ‘investigator calculated’ and ‘TB programme staff recorded’ BMI differed by more than one kg/m^2^. Among those screened, we also derived the extent of missing or illegal data.

### 2.6. Statistics

We performed crude comparisons of proportions using the chi square test. We used modified Poisson regression with robust variance estimates to determine the factors associated with screening for severe illness (‘yes’). We summarised the association using adjusted prevalence ratios. We included factors with crude *p* < 0.05 in the regression model after ruling out multi-collinearity (district was also included as a potential confounder).

## 3. Results

### 3.1. Baseline Characteristics of Notified Patients

Of 3020 study participants, the number notified was as high as 682 in district-3 and as low as 62 in district-4 (see Figure 1). 

Sixty-seven percent were diagnosed in district hospitals or teaching hospitals and bank details (used for direct benefit transfer) were available for 78%. While same day notification happened in 72% patients, 5.5% were not started on treatment and 17.3% were transferred out of district after diagnosis for continuation of treatment (see Table 1).

### 3.2. Feasibility Indicators for Screening of Severe Illness in Routine Settings

#### 3.2.1. Coverage of Screening and Factors Associated

Of 3020 patients, 50.6% (n = 1531) were screened, this varied across districts (13–90%) (see Figure 1). 

On adjusted analysis, people living with HIV, with no bank details, notified from district hospitals or medical colleges, notified two weeks or more after diagnosis, not initiated on treatment or transferred out of district were less likely to be screened (see Table 2).

#### 3.2.2. Delays in Screening

The median time interval from notification to screening at PHI was two (IQR: 0, 8) days, from screening at PHI to uploading the details in the mobile application was zero (IQR: 0, 1) days and from notification to uploading the details in the mobile application was five (IQR: 1, 12) days. The latter varied across districts and was as short as one day and as long as 10 days.

#### 3.2.3. Errors in Data Collected and Missing Data

Of 1531 screened, data was collected for all the indicators of our screening tool in 88% patients. Errors in calculation of BMI were seen in 8% patients. Respiratory rate data was missing in 7% of patients and oxygen saturation data in 6% patients (see Table 3).

### 3.3. Burden of ‘High Risk of Severe Illness’

The data presented here is based on investigator-derived calculations using the primary data collected by TB programme staff. The distribution of weight, BMI, respiratory rate (RR) and oxygen saturation is given in Table 4. The burden of ‘high risk of severe illness’ was 35.1% (95% CI: 32.8, 37.6) (see Table 5). One-fourth (130/538) of people with ‘high risk of severe illness’ were admitted at the time of screening. In the screening tool, for BMI, if we replace ‘≤’ with ‘<’ sign, the burden would be 34.7% (95% CI: 32.4, 37.1).

### 3.4. Contribution of Various Indicators to ‘High Risk of Severe Illness’

The contribution of very severe undernutrition indicators, respiratory insufficiency indicators and inability to stand without support towards total burden was 34%, 64% and 28%, respectively (see Figure 2) and there was some overlap (see Figure 3). People classified based on the presence of very severe undernutrition indicators only (any one) contributed to 20% (108/538) of the total burden. Similarly, independent contribution of respiratory insufficiency indicators (any one) to the total burden was 45% (244/538) and inability to stand without support was 13% (71/538). Five percent patients (27/538) were positive for all three categories of indicators.

## 4. Discussion

### 4.1. Strengths and Limitations

To the best of our knowledge, this is the first ever experience on screening for severely ill people with TB in routine programme settings from India and globally. Karnataka State has taken the initiative and this study paves the way for other states in India and other high burden countries to follow suit. This is the first large cohort from a TB programme setting in India with data regarding BMI, vitals and performance status.

A study from south India (2018–19) recommended differentiated TB care (in the form of intensive treatment support) for people with any one of the following risk factors: age ≥ 60 years, living alone, HIV, diabetes, previous treatment, drug-resistant TB, regular alcohol consumption and undernutrition (weight less than 43 kg for men and <38 kg for women) [27]. BMI was not used as height was not measured. The prevalence of at least one risk factor was 30% and these people had 3.3 times higher odds of death when compared to those with no risk factor. Assessment of risk was done at around two months post-treatment. Only those who were assessed were included and by two months most of the deaths would have occurred [27]. The indicators suggested by us in the screening tool are known risk factors for death [14,23,24,25,26] and are not dependent on laboratory assessment. Our experience shows that these can be measured within few days of diagnosis/notification and are therefore actionable to prevent deaths.

There were three major limitations. First, among those with ‘high risk of severe illness’, other than those who were admitted at the time of screening, we were not able to confirm the presence of severe illness. People with ‘high risk of severe illness’ were not systematically referred to higher facility for clinical evaluation due to lack of guidance/policy from the state at the time of our study. Future studies should assess the validity of this screening tool including how many severely ill patients are missed during screening. Second, due to the ongoing COVID-19 pandemic, we were not able to conduct in-person trainings and perform a reliability assessment of the measurements done by TB programme staff. However, we feel the observed data quality was acceptable for a programme setting. Third, COVID-19 response related lockdown resulted in TB under-detection and low coverage of screening. We are not sure if the burden of illness found in those screened can be extrapolated to those who were not screened. Due to prevailing COVID-19 and its response, it is possible that only those who were severely ill sought care. Hence, we may be over-estimating the burden when compared to pre-COVID-19 period. Even if we assume a best-case scenario (i.e., all those who were not screened or not notified did not have ‘high risk of severe illness’), the burden would still be 13% (data not shown) and this is of public health importance.

### 4.2. Key Findings

The burden of ‘high risk of severe illness’ accounted for one-third of all patients. All three categories of indicators that contributed to the burden need to be retained in the screening tool. They had significant independent contribution to the burden. The proportion of patients with undernutrition (BMI < 18.5 kg/m^2^, 55% in men and 54% in women) is lower than the findings from Chhattisgarh (87% in men and 93% in women) [28]. Similarly, the proportion of patients with severe undernutrition (BMI < 16 kg/m^2^, 29% in men and 32% in women) is also lower when compared to patients from Chhattisgarh (50% in men and 66% in women) [28].

Feasibility indicators in favour of screening were the short time interval for screening, all indicators being collected in most of the patients and acceptable data quality for a programme setting. The feasibility indicator not in favour of screening was that half of the patients were not screened. While some districts performed exceedingly well despite COVID-19, other districts performed poorly. We are unclear about the facilitators and barriers of screening and this needs further investigation using qualitative research methods.

The screening coverage was low in patients notified from district level or teaching hospitals and we speculate this may be related to high patient load in these settings. Poor coverage of screening among people with HIV requires urgent attention. In Karnataka, high HIV prevalence among people with TB has been documented as one of the reasons for high case fatality [27]. People without bank details were less likely to be screened for severe illness. Since such patients are more likely to be from marginalised and vulnerable sections of the population (for example migrant populations), they are more likely to be severely ill [29,30]. Non-initiation of treatment, delays in notification and transfer outs were associated with low coverage of screening. These can be addressed by ensuring screening happens immediately at diagnosis without waiting for notification.

### 4.3. Recommendations

We recommend that screening be done routinely in the programme and ‘high risk of severe illness’ (yes/no) be recorded as a baseline characteristic in NIKSHAY. Once the TB services return to pre-COVID level and with the political and administrative will to attain the 2030 World Health Organization End TB targets by 2025 [18], we can attain higher coverage of screening. The 2021 guidance recommending severity assessment for all adults with TB may provide the necessary stimulus in making this happen [15]. Errors in BMI calculation may be reduced by the use of N-TB application (endorsed by NTEP) [31] and/or using an auto-generated field for BMI in the mobile application. Similarly, an auto-generated field may be considered for ‘high risk of severe illness’.

Along with routine capture of ‘high risk of severe illness’, we recommend the following two interventions to reduce TB deaths. First, interventions to improve case finding and early diagnosis and treatment. This can be tracked through change (reduction) in proportion of people with TB with ‘high risk of severe illness’ over time.

Second, those with ‘high risk of severe illness’ may be prioritized for referral to higher facilities for comprehensive clinical evaluation and assessment for inpatient care. The modality of implemention should be assessed by another operational research by capturing additional variables: referral (yes/no), comprehensive clinical evaluation (yes/no), eligible for inpatient care (yes/no) and eventual inpatient care (yes/no). Once implemented systematically, the programme may track the change in TB mortality indicators over time.

The TB and TB-comorbidity related codes available in the Ayushmann Bharat scheme (a public insurance scheme for the poor) may be utilized to provide insurance cover during inpatient care to reduce catastrophic costs [32]. Inpatient care should also focus on therapeutic nutritional care.

## 5. Conclusions

We have described our experience from Karnataka (South India) where para-medical programme staff screened adults with TB for ‘high risk of severe illness’, defined using indicators of very severe undernutrition, abnormal vital signs and poor performance status. The possible high burden in our setting emphasizes the importance of the recently released national guidance on early detection and referral of people with TB who are severely ill for inpatient care. We found the tool feasible to use. The coverage of screening was low and might be related to the COVID-19 pandemic related burden on the health system. We recommend that ‘high risk of severe illness (yes/no)’ be included in routine recording and reporting. Those screened as ‘high risk of severe illness’ may undergo comprehensive clinical evaluation at a higher facility and provided inpatient care if eligible. Efforts like these are urgently needed for ending TB deaths.

## Figures and Tables

**Figure 1 tropicalmed-06-00102-f001:**
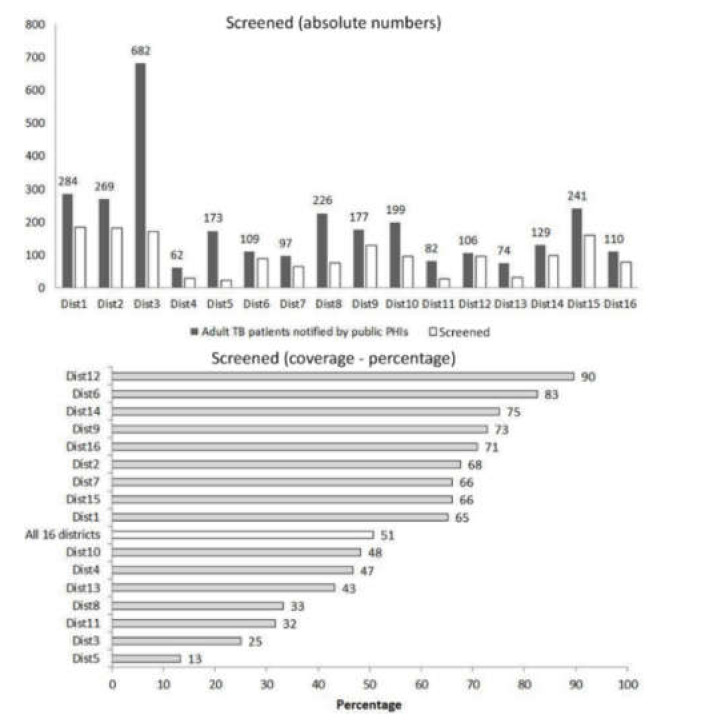
Screening ^(^^$)^ for ‘high risk of severe illness’ (absolute numbers and percentage) at notification among adults (≥15 years) with TB (without known drug-resistant disease at diagnosis) from public health facilities of 16 districts in Karnataka, India, 15 October to 30 November 2020 (n = 3020). TB—tuberculosis; ^(^^$)^ defined as filling and syncing of screening details in a mobile application irrespective of the extent of missing data.

**Figure 2 tropicalmed-06-00102-f002:**
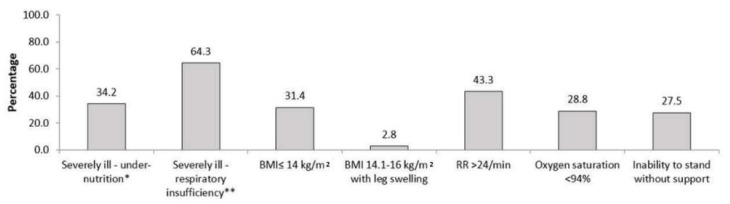
Contribution ^(^^$)^ of individual indicators to ‘high risk of severe illness’ at notification among adults (≥15 years) with TB (without known drug-resistant disease at diagnosis) from public health facilities of 16 districts in Karnataka, India, 15 October to 30 November 2020 (n = 538) ^(@)^. * Presence of any one indicator–BMI ≤ 14 kg/m^2^, BMI ≤ 14.1–16 kg/m^2^ with leg swelling; ** Presence of any one indicator–respiratory rate > 24/min, oxygen saturation < 94%; TB—tuberculosis, BMI—body mass index, ^(^^$)^ Percentages will add up to more than 100%, more than one indicator may be present in an individual; ^(@)^ of 3020 people with TB, a total of 1531 were screened, of 1531, a total of 538 had ‘high risk of severe illness’.

**Figure 3 tropicalmed-06-00102-f003:**
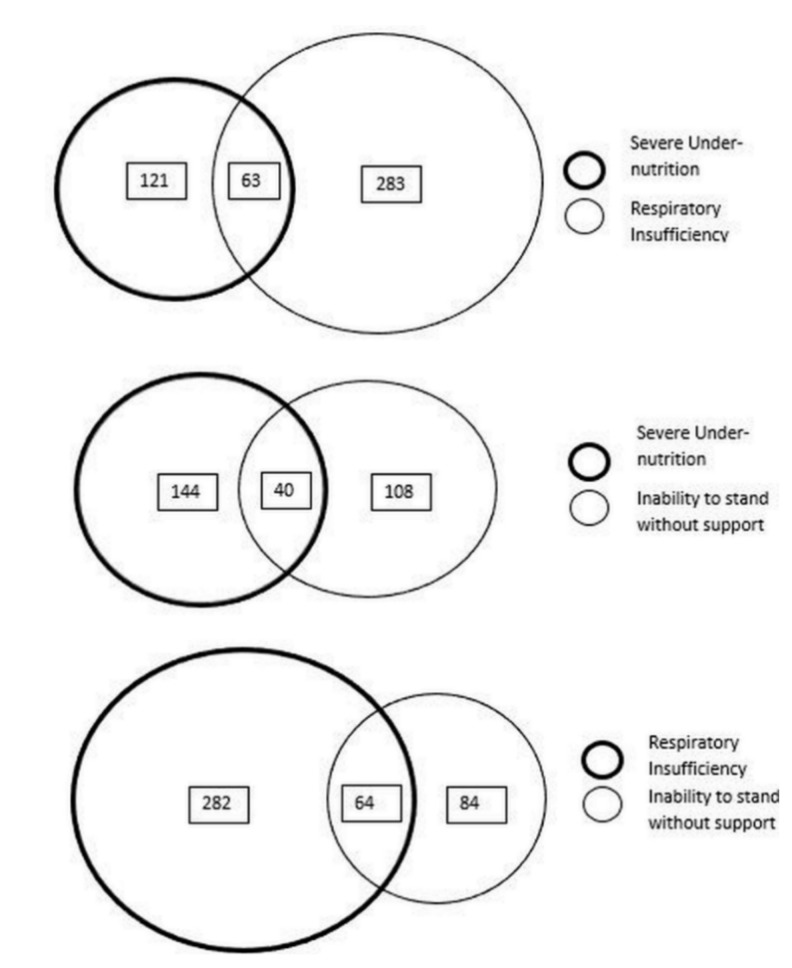
Overlapping ^(^^$)^ of indicators among adults (≥15 years) with TB (without known drug-resistant disease at diagnosis) classified as ‘high risk of severe illness’ at notification from public health facilities of 16 districts in Karnataka, India, 15 October to 30 November 2020 (n = 538) ^(@)^. ^(^^$)^ Size of circles is only representational and not directly proportional to the numbers under each category, value mentioned in boxes indicates the number with ‘high risk of severe illness’; ^(@)^ of 3020 people with TB, a total of 1531 were screened, of 1531, a total of 538 had ‘high risk of severe illness’.

**Table 1 tropicalmed-06-00102-t001:** Baseline characteristics of adults (≥15 years) with TB (without known drug-resistant disease at diagnosis) notified from public health facilities of 16 districts in Karnataka, India, 15 October to 30 November 2020 (n = 3020).

Characteristics *		n	(%)
Total		3020	(100.0)
Demographic characteristics			
Age in years			
	15–24	476	(15.8)
	25–34	611	(20.2)
	35–44	619	(20.5)
	45–54	575	(19.0)
	55–64	416	(13.8)
	≥65	323	(10.7)
Gender			
	Men	2039	(67.5)
	Women	981	(32.5)
**Clinical characteristics**			
Test used for diagnosis			
	Xpert MTB/RIF	1423	(47.1)
	TrueNat MTB/RIF	190	(6.3)
	LPA	17	(0.6)
	Microscopy	557	(18.4)
	Culture	9	(0.3)
	Chest Radiograph	272	(9.0)
	Others	552	(19.3)
Bacteriological confirmation (yes)		2284	(75.6)
Site			
	Pulmonary	2186	(72.4)
	Extrapulmonary	667	(22.1)
	Missing	167	(5.5)
Previous treatment (yes)		474	(15.7)
HIV			
	Positive	225	(7.5)
	Negative	2531	(83.8)
	Unknown	264	(8.7)
DM			
	Yes	505	(16.7)
	No	2086	(69.1)
	Unknown	429	(14.2)
**Health system characteristics**			
Bank details available with programme (yes)		2353	(77.9)
Peripheral Health Institute–notification facility			
	District/Teaching hospital	1942	(66.6)
	Sub-district level hospital	769	(26.4)
	Primary health centre	206	(7.1)
Days to notify from diagnosis			
	Within a day	2160	(71.5)
	1–6 days	632	(20.9)
	7–13	113	(3.7)
	14–27	73	(2.4)
	≥28	42	(1.4)
Treatment not started		167	(5.5)
Transferred out of district (yes)		521	(17.3)

TB—tuberculosis, LPA line probe assay, HIV—human immunodeficiency virus, DM—Diabetes mellitus; * source is the routinely collected baseline data in NIKSHAY updated as on 15 December 2020.

**Table 2 tropicalmed-06-00102-t002:** Factors associated with screening for ‘high risk of severe illness ^@^ at notification among adults (≥15 years) with TB (without known drug-resistant disease at diagnosis) from public health facilities of 16 districts in Karnataka, India, 15 October to 30 November 2020 (n = 3020).

Factors *		Total	Screened (Yes)		PR	(95%CI)	aPR **	(95%CI)
			n	(%)				
Total		3020	1531	(50.7)				
Age in years								
	15–24	476	234	(49.2)	Ref		^#^	
	25–34	611	305	(49.9)	1.02	(0.90, 1.15)		
	35–44	619	321	(51.9)	1.05	(0.94, 1.19)		
	45–54	575	290	(50.4)	1.03	(0.91, 1.16)		
	55–64	416	207	(49.8)	1.01	(0.89, 1.16)		
	≥65	323	174	(53.9)	1.10	(0.96, 1.26)		
Gender								
	Men	2039	1045	(51.3)	Ref		^#^	
	Women	981	486	(49.5)	0.97	(0.90, 1.04)		
Test used for diagnosis								
	Rapid molecular tests	1630	789	(48.4)	Ref		Ref	
	Microscopy/Culture	566	294	(51.9)	1.07	(0.98, 1.18)	1.07	(0.97, 1.18)
	Chest Radiograph	272	168	(61.8)	1.28	(1.15, 1.42) ^	1.09	(0.98, 1.20)
	Others	552	280	(50.7)	1.05	(0.95, 1.15)	1.16	(1.05, 1.27) ^
Bacteriological confirmation								
	Yes	2284	1135	(49.7)	Ref		^#^	
	No	736	396	(53.8)	1.08	(1.00, 1.17)		
Site of TB								
	Pulmonary	2186	1210	(55.4)	Ref		^&^	
	Extrapulmonary	667	318	(47.4)	0.86	(0.79, 0.94) ^		
	Missing	167	3	(1.8)	0.03	(0.01, 0.10) ^		
Previous treatment								
	Yes	474	248	(52.3)	1.04	(0.94, 1.14)	^#^	
	No	2546	1283	(50.4)	Ref			
HIV								
	Positive	225	109	(48.4)	0.90	(0.79, 1.04)	0.86	(0.76, 0.98) ^
	Negative	2531	1358	(53.7)	Ref			
	Unknown	264	64	(24.2)	0.45	(0.36, 0.56) ^	0.85	(0.69, 1.05)
DM								
	Positive	505	274	(54.3)	0.99	(0.91, 1.09)	0.99	(0.91, 1.07)
	Negative	2086	1138	(54.6)	Ref			
	Unknown	429	119	(27.7)	0.51	(0.43, 0.60) ^	0.82	(0.71, 0.95) ^
Bank details available								
	Yes	2353	1281	(54.4)	Ref			
	No	667	250	(37.5)	0.69	(0.62, 0.76) ^	0.87	(0.79, 0.95) ^
Peripheral Health Institute—Notification facility								
	District/Teaching hospital	1942	847	(43.6)	0.65	(0.60, 0.70) ^	0.87	(0.81, 0.94) ^
	Sub-district level hospital	769	517	(67.2)	Ref			
	Primary health Centre	206	100	(48.5)	0.72	(0.62, 0.84) ^	1.11	(0.95, 1.29)
Days to notify from diagnosis								
	Within a day	2160	1098	(50.8)	Ref			
	1–6 day	632	352	(55.7)	1.10	(1.01, 1.19) ^	1.07	(0.99, 1.16)
	7–13	113	54	(47.8)	0.94	(0.77, 1.14)	0.98	(0.82, 1.18)
	14–27	73	22	(30.1)	0.59	(0.42, 0.84) ^	0.71	(0.52, 0.98) ^
	≥28	42	5	(11.9)	0.23	(0.10, 0.53) ^	0.31	(0.14, 0.68) ^
Treatment started								
	Yes	2853	1528	(53.6)	Ref			
	No	167	3	(1.8)	0.03	(0.01, 0.10) ^	0.04	(0.01, 0.17) ^
Transferred out of district								
	Yes	521	156	(29.9)	0.54	(0.47, 0.62) ^	0.70	(0.62, 0.80) ^
	No	2499	1375	(55.0)	Ref			

TB—tuberculosis, PTB—pulmonary TB, EPTB—extrapulmonary TB, LPA—line probe assay, HIV—human immunodeficiency virus, DM—Diabetes mellitus; PR—crude prevalence ratio, aPR—adjusted PR; ^@^ defined as filling and syncing of screening details in a mobile application irrespective of the extent of missing data; * source is the routinely collected baseline data in NIKSHAY updated as on 15 December 2020; ** modified Poisson regression with robust variance estimation, results have been adjusted for district;^#^ Variables with crude *p* value ≥ 0.05 (Chi square test) were not included in the adjusted analysis;^&^ excluded because of variance inflation factor >10; ^ *p* < 0.05.

**Table 3 tropicalmed-06-00102-t003:** Missing data and errors in data collected during screening for ‘high risk of severe illness’ at notification among adults (≥15 years) with TB (without known drug-resistant disease at diagnosis) from public health facilities of 16 districts in Karnataka, India, 15 October to 30 November 2020 (n = 1531)^@^.

	Total Screened	
	n	(%)
Total	1531	(100.0)
Data on all indicators collected	1354	(88.4)
Illegal entry or missing weight or height (missing BM)	7	(0.5)
Errors in BMI calculation	126	(8.3)
Instances where BMI ≤ 14.0 and able to stand without support	133	(8.7)
Missing details on leg swelling	0	(0)
Missing details on respiratory rate	99	(6.5)
Missing details on oxygen saturation	88	(5.7)
Missing details on ability to stand without support	0	(0)

TB—tuberculosis, BMI—body mass index (kg/m^2^); ^@^ of 3020 patients, a total of 1531 (50.7%) were screened.

**Table 4 tropicalmed-06-00102-t004:** Distribution of body mass index, respiratory rate and oxygen saturation at notification among adults (≥15 years) with TB (without known drug-resistant disease at diagnosis) screened for ‘high risk of severe illness’ from public health facilities of 16 districts of Karnataka, India, 15 October to 30 November 2020 (n = 1531) ^@^.

		Total		Men		Women	
Characteristics		n	(%)	n	(%)	n	(%)
Total		1531	(100.0)	1045	(100.0)	486	(100.0)
Weight (kg)							
	<30	33	(2.2)	9	(0.9)	24	(4.9)
	30–44	594	(38.8)	357	(34.2)	237	(48.8)
	45–59	674	(44.0)	516	(49.4)	158	(32.5)
	≥60	223	(14.6)	160	(15.3)	63	(13.0)
	Missing	7	(0.5)	3	(0.3)	4	(0.8)
	*Mean (SD)*	*47.8*	*(11.9)*	*49.1*	*(11.4)*	*45.0*	*(12.4)*
Body mass index (kg/m^2^)							
	≤14.0	169	(11.0)	106	(10.1)	63	(13.0)
	14.1–16.0	288	(18.8)	198	(18.9)	90	(18.5)
	16.1–18.4	379	(24.8)	273	(26.1)	106	(21.8)
	≥18.5 ^$^	688	(44.9)	465	(44.5)	223	(45.9)
	Missing	7	(0.5)	3	(0.3)	4	(0.8)
	*Mean (SD)*	*18.7*	*(4.4)*	*18.5*	*(4.1)*	*19.0*	*(5.1)*
Respiratory rate per minute							
	<18	240	(15.7)	158	(15.1)	82	(16.9)
	18–24	959	(62.6)	655	(62.7)	304	(62.6)
	25–30	120	(7.8)	77	(7.4)	43	(8.8)
	>30	113	(7.4)	86	(8.2)	27	(5.6)
	Missing	99	(6.5)	69	(6.6)	30	(6.2)
Oxygen saturation (%)							
	≥94	1288	(84.1)	870	(83.3)	418	(86.0)
	90–93	123	(8.0)	88	(8.4)	35	(7.2)
	85-89	14	(0.9)	12	(1.1)	2	(0.4)
	<85	18	(1.2)	15	(1.4)	3	(0.6)
	Missing	88	(5.7)	60	(5.7)	28	(5.8)

TB—tuberculosis, ^@^ of 3020 patients, a total of 1531 were screened; ^$^ 162 were overweight (BMI 23.0–27.4) and 61 obese (BMI ≥ 27.5).

**Table 5 tropicalmed-06-00102-t005:** Burden of ‘high risk of severe illness’ at notification using our screening among adults (≥15 years) with TB (without known drug-resistant disease at diagnosis) from public health facilities of 16 districts in Karnataka, India, 15 October to 30 November 2020 (n = 1531) ^@^.

Criteria	n	%	(95% CI)
Using the screening criteria	538	35.1	(32.8, 37.6)
Using BMI ≤ 14	169	11.0	(9.6, 12.7)
Using BMI 14–16 with leg swelling	15	1.0	(0.6, 1.6)
Using RR > 24/min	233	15.2	(13.5, 17.1)
Using oxygen saturation < 94%	155	10.1	(8.7, 11.7)
Inability to stand without support	148	9.7	(8.3, 11.2)
Very severe undernutrition related indicator (any one)	184	12.0	(10.5, 13.7)
Respiratory insufficiency related indicator (any one)	346	22.6	(20.6, 24.8)

TB—tuberculosis, BMI—body mass index (kg/m^2^), RR—respiratory rate; ^@^ of 3020 people with TB, a total of 1531 were screened.

## Data Availability

The study data used has been shared as Appendix A (Appendix A—study dataset and codebook in MS Excel format).

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
