# Peer review of "Screening People with Tuberculosis for High Risk of Severe Illness at Notification: Programmatic Experience from Karnataka, India"

_tropicalmed, 2021, doi:10.3390/tropicalmed6020102_

Round 1
Reviewer 1 Report
This is a good and interesting paper.
Author Response
Thank you for the constructive comment
Reviewer 2 Report
This is a nice study, quite well written and laid out. It would be improved with addition of another table detailing the results of the clinical parameters collected across the various severe or not severe disease groups, to provide a baseline for other future studies and which would be useful to clinicians in the field, if these are available and can be collated. I think this study is of sufficient novelty to merit publication and the demographic information will be extremely useful for other workers in the field.
The English language is very good, I have minor comments on that, which I have given in an annotated hard copy attached. There are some paragraphs where the meaning is not clear and which need revision and where additional references need to be given. One additional comment is to be consistent with headings where the initial letter should be in capitals with the exception of and, or etc. With abbreviations - the abbreviation of number is given as either N or n and should be given as either No. if using capitals or in lowercase format given as n and should be consistent throughout the document. These are minor points but need tidying up.

Author Response
Reviewer
This is a nice study, quite well written and laid out. It would be improved with addition of another table detailing the results of the clinical parameters collected across the various severe or not severe disease groups, to provide a baseline for other future studies and which would be useful to clinicians in the field, if these are available and can be collated. I think this study is of sufficient novelty to merit publication and the demographic information will be extremely useful for other workers in the field.
Authors
Thank you very much for the constructive comments. This was an operational research involving primary and secondary data. Secondary data involved all the baseline characteristics collected by the TB programme for each TB patient. Primary data involved severe illness screening related data (this is not routinely collected). We have shared the available data collected. Unfortunately, we do not have the clinical parameters for all the patients. As discussed in the introduction section, that there is lack of diagnostic infrastructure at peripheral health instituions for comprehensive clinical evaluation for all the TB patients. Hence, we suggested addition of these easy to collect and interpret variables that can be used by para-medical TB programme staff and these do not require the need for laboratory, radiological or clinical evaluation.
Reviewer
The English language is very good, I have minor comments on that, which I have given in an annotated hard copy attached. There are some paragraphs where the meaning is not clear and which need revision and where additional references need to be given. One additional comment is to be consistent with headings where the initial letter should be in capitals with the exception of and, or etc. With abbreviations - the abbreviation of number is given as either N or n and should be given as either No. if using capitals or in lowercase format given as n and should be consistent throughout the document. These are minor points but need tidying up.
Authors
We have noted the comment regarding the heading and the abbreviation to be used for number. We have added references as advised. Edits have been made in the revised manuscript. Thank you.
We have noted the attached copy and addressed these comments. Most of the minor edits have been incorporated in the revised manuscript. Detailed explanation for some of the comments may be found below. We have also mentioned reasons below for not considering some edits suggested by the reviewer.
We have retained undernutrition, instead of the suggested malnutrition. Malnutrition may mean under or over nutrition. Here specifically undernutrition was our focus
We have retained ‘routine programme settings’ and decided not to add ‘surveillance’ between programme and settings.
14:100 is the estimated case fatality ratio globally. 17:100 is the estimated case fatality ratio for India
Regarding some content from first three paragraph to be included in introduction, the authors reviewed the manuscript again and felt the content in first three paragraphs of settings is more suitable for settings. We think the introduction flows well in the current format. We hope this is fine.
Bank details are captured and this is recorded as yes/no in the NIKSHAY database. This is then used to transfer 500 Indian rupees per month during TB treatment for all the patients. Non-availability of bank accounts is a proxy indicator for poor socio-economic status. We have discussed the relevance of this factor in the discussion section. People without bank details were less likely to be screened for severe illness. Since such patients are more likely to be from marginalised and vulnerable sections of the population (for example migrant populations), they are more likely to be severely ill. Please see line 343-46 of revised manuscript with track changes. We have clarified on first use in narrative text that bank account is used for direct benefit transfer.
Some comments regarding tables (within one page), is not within our control. Journal guidelines suggest that they should be inserted at the place of citing in narrative text. Same applies to the layout.
Venn diagram – We did think of a 3-way diagram, but then realised it would be complicated and it will be difficult to depict the number in each area.
Inpatient care should focus on therapeutic nutritional care (statement in discussion). This based on the finding that a large contribution to severe illness is made by very severe under-nutrition. As this is based on our study finding, this does not require reference.
Regarding a space to left of paragraphs, we have followed the formatting as per the template provided by the journal
Reviewer 3 Report
In this study, the authors study programmatic assessment of clinical markers of TB severity in 16 districts of one South Indian state. They develop a multivariable statistical model to identify predictors of persons successfully screened for markers of disease severity. They aslo describe the pattern of severity markers and indicate which clinical markers were prone to being collected erroneously.
The authors rightly point out that TB programs should be paying attention to markers of severity and referring patients to inpatient care accordingly.
Major point:
One key concern I had was regarding the specific set of disease severity markers they selected for this study. Although eminently rational and pragmatic, the value of the set of disease markers chosen in this study is of unclear value. The article refers the reader to reference 12 to understand the selection of the specific markers.
In that reference Bhargava and Bhargava write:
"A triage tool proposed by the national TB program in India includes one or more of the following: (BMI ≤ 14 kg/m2 or BMI ≤ 16 kg/m2 with pedal edema, or MUAC ≤ 19 cm, signs of respiratory insufficiency (assessed clinically by breathlessness or respiratory rate > 24/min or an oxygen saturation on pulse oximetry < 94%), and an inability to stand as features which indicate high risk and a need for admission. This has not been validated independently although the inability to walk unaided, and a low BMI were independent predictors of mortality in a cohort of seriously ill HIV infected patients with suspected TB [71]. Tachycardia, tachypnea and inability to walk unaided are also danger signs suggested by WHO to identify seriously ill patients with HIV disease [72]."
From this I conclude, that the triage tool for disease severity suggested here is not a validated instrument for disease severity. This is a key limitation as the reader may well ask why it matters that a program is able to collect these data if they haven't been proven to correlate with severe disease that poses harm in the absence of hospitalization. At the very least, this should be acknowledged as a major study limitation.
Minor point:
For their regression, the authors included all variables with crude p <0.05. I would have expected them to include all variables at a significance level of 0.15 or at least 0.10.
Overall, this is a meaningful paper and the authors' findings that it is possible to configue India's national TB program to triage patients and refer the most vulnerable patients to inpatient care.
Author Response
Reviewer
In this study, the authors study programmatic assessment of clinical markers of TB severity in 16 districts of one South Indian state. They develop a multivariable statistical model to identify predictors of persons successfully screened for markers of disease severity. They aslo describe the pattern of severity markers and indicate which clinical markers were prone to being collected erroneously.
The authors rightly point out that TB programs should be paying attention to markers of severity and referring patients to inpatient care accordingly.
Authors
Thank you for the constructive comments
Reviewer
Major point:
One key concern I had was regarding the specific set of disease severity markers they selected for this study. Although eminently rational and pragmatic, the value of the set of disease markers chosen in this study is of unclear value. The article refers the reader to reference 12 to understand the selection of the specific markers.
In that reference Bhargava and Bhargava write:
"A triage tool proposed by the national TB program in India includes one or more of the following: (BMI ≤ 14 kg/m2 or BMI ≤ 16 kg/m2 with pedal edema, or MUAC ≤ 19 cm, signs of respiratory insufficiency (assessed clinically by breathlessness or respiratory rate > 24/min or an oxygen saturation on pulse oximetry < 94%), and an inability to stand as features which indicate high risk and a need for admission. This has not been validated independently although the inability to walk unaided, and a low BMI were independent predictors of mortality in a cohort of seriously ill HIV infected patients with suspected TB [71]. Tachycardia, tachypnea and inability to walk unaided are also danger signs suggested by WHO to identify seriously ill patients with HIV disease [72]."
From this I conclude, that the triage tool for disease severity suggested here is not a validated instrument for disease severity. This is a key limitation as the reader may well ask why it matters that a program is able to collect these data if they haven't been proven to correlate with severe disease that poses harm in the absence of hospitalization. At the very least, this should be acknowledged as a major study limitation.
Authors
Thank you very much. Regarding validation, we have mentioned this as a study limitation. We provided a recommendation that future studies should validate this tool esp how many severely ill patients is it missing (extent of false negative). Please refer to last line of abstract and 308-10 lines of revised manuscript with track changes (limitations section)
Last line of abstract
“Future studies should assess the validity (especially sensitivity in picking severely ill patients) of this screening tool.”
Lines 308-10 of revised manuscript with track changes (limitations section)
“Future studies should assess the validity of this screening tool including how many severely ill patients are missed during screening”
Reviewer
Minor point:
For their regression, the authors included all variables with crude p <0.05. I would have expected them to include all variables at a significance level of 0.15 or at least 0.10.
Authors
Thank you for the comment. Based on the reviewer suggestion, we repeated the analysis after including all variables at a significance level of 0.15. We repeated this exercise using a significance level of 0.10. There was no change in factors that were significantly associated. Hence, we would not like to make any changes to the regression as of now. We hope this is fine.
Reviewer
Overall, this is a meaningful paper and the authors' findings that it is possible to configue India's national TB program to triage patients and refer the most vulnerable patients to inpatient care
Authors
Thank you for the constructive comments.